# Post cesarean section surgical site infection and associated factors among women who delivered in public hospitals in Harar city, Eastern Ethiopia: A hospital-based analytic cross-sectional study

**Tsegaw Alemye[1], Lemessa Oljira[2], Gelana Fekadu[3], Melkamu Merid Mengesha[4]***

**1** Department of Midwifery, Mizan-Aman Health Sciences College, Mizan-Aman, Ethiopia, **2** School of Public Health, College of Health and Medical Sciences, Haramaya University, Harar, Ethiopia, **3** School of Nursing and Midwifery, College of Health and Medical Sciences, Haramaya University, Harar, Ethiopia, **4** School of Public Health, College of Medicine and Health Sciences, Arba Minch University, Arba Minch, Ethiopia

* melkamumrd@gmail.com

**Data Availability Statement:** All relevant data are within the manuscript and its Supporting Information files.

## Abstract

### Background

Cesarean section (CS) is often complicated by surgical site infection (SSI) that may happen to a woman within 30 days after the operation. This study was conducted to estimate the prevalence of SSI and identify the factors associated with SSI.

### Methods

A hospital-based analytic cross-sectional study was conducted based on the review of medical records of 1069 women who underwent CS in two public hospitals in Harar city. The post-CS SSI is defined when it occurred within 30 days after the CS procedure. Factors associated with SSI were identified using a multivariable binary logistic regression analysis. The analysis outputs are presented using an adjusted odds ratio (aOR) with a corresponding 95% confidence interval (CI). All statistical tests are defined as statistically significant at P-values<0.05.

### Results

The prevalence of SSI was 12.3% (95% confidence interval (CI): 10.4, 14.4). Emergency-CS was conducted for 75.9% (95% CI: 73.2, 78.3) of the women and 13.2% (95% CI: 11.3, 15.4) had at least one co-morbid condition. On presentation, 21.7% (95% CI: 19.3, 24.3) of women had rupture of membrane (ROM). Factors significantly and positively associated with post-CS SSI include general anesthesia (aOR = 2.0, 95%CI: 1.10, 2.90), ROM (aOR = 2.27, 95%CI: 1.02, 3.52), hospital stay for over 7 days after operation (aOR = 3.57, 95%CI: 1.91, 5.21), and blood transfusion (aOR = 4.2, 95%CI: 2.35, 6.08).

**Funding:** TA-received grant from Haramaya University for data collection. The funders had no role in study design, data collection and analysis, decision to publish, or preparation of the manuscript.

**Competing interests:** The authors have declared that no competing interests exist.

## Conclusion

The prevalence of post-CS SSI was relatively high in the study settings. Screening for pre-operative anemia and appropriate correction before surgery, selection of the type of anesthesia, close follow-up to avoid unnecessary prolonged hospitalization, and careful assessment of membrane status should be considered to avoid preventable SSI and maternal morbidity.

## Introduction

Cesarean section (CS) is a lifesaving operative technique by which a fetus, the placenta, and membranes are delivered through an abdominal and uterine incision [1]. Even though it is a lifesaving procedure, it can also carry a significant risk both for the mother and the newborn including severe acute maternal morbidity, post-partum infection or death [2–4]. Surgical site infection (SSI) is one among such risks that may happen to a women after a CS procedure. It is an infection that happens at the incision/operative site within 30 days of the post-surgical procedure [5].

Globally, SSI is the second most reported health-care associated infections (HAI) accounting for 19.6% of the HAIs [6]. In a report of the burden of HAI in developing countries, Allegranzi et al reported a pooled cumulative incidence of SSIs of 11.8 per 100 patients undergoing surgical procedures and 5.6 per 100 surgical procedures [7]. A systematic review of HAI in African hospitals by Nejad et al reported a cumulative incidence that ranged from 2.5% to 30.9% [8]. Particular to Ethiopia, a meta-analysis by Adane et al. reported an 8.8% pooled estimate of the prevalence of post-CS SSI [9].

Research evidences reported from different settings indicated that several factors contribute to SSIs including a preexisting comorbid conditions, age, duration of operation, anemia, frequency of manual vaginal examination, and inappropriate antibiotic prophylaxis [10–12]. Hence, identification of such risk factors that increase SSIs may help to design interventions that consider the context where such operations are performed.

Clinical practice recommendations for the prevention of SSI, among others, include prophylactic antibiotic use, chlorhexidine skin preparation, hair removal using clippers, and vaginal cleansing using povidone-iodine [13]. Despite these infection prevention efforts, no significant decline is achieved and, post-CS SSI remains a public health problem [14]. The purpose of this study was to assess the magnitude and factors associated with post-CS SSI among mothers who underwent cesarean section at public Hospitals in Harar city, Eastern Ethiopia.

## Methods and materials

### Study area and period

A fifteen months data, from October 11, 2018 to December 31, 2019, was extracted from patient records during March 1 to March 15, 2020 from two public hospitals in Harar city, namely Jugol General Hospital and Hiwot Fana Specialized University Hospital (HFSUH), Eastern Ethiopia. Harar city is located at 526 km towards the east of Addis Ababa, the Ethiopian capital. According to the 2007 census conducted by the Central Statistical Agency, the total population of the region is estimated to be 183, 415 [15]. Ethiopia's health service is structured into a three tier system: primary healthcare unit (health post (serve 3000–5000 people), health center (serve 15000–40,000 people), and primary hospital (serve 60,000–100,000

people)), secondary level healthcare (general hospital: serve 1.0–1.5 million people), and tertiary level healthcare (specialized hospital: 3.0–5.0 million people). When this study was conducted, there were two public hospitals (general and specialized hospitals), two private general hospitals, one police-, and one non-governmental fistula hospital in the region. In addition to these hospitals in the city, there were 29 private clinics, 26 health posts, eight health centers, and one regional laboratory. In the two hospitals, in terms of human resource to provide CS services, there were 7 gynecology and obstetrics senior specialists, 33 residents, and 3 integrated emergency surgical officers (only in Jugol General Hospital).

## Study design

Hospital-based analytic cross-sectional audit of patient records (from October 11, 2018 to December 31, 2019) was conducted to estimate the magnitude of post-CS SSI and identify the associated factors among women who underwent CS in two public hospitals.

## Population

The source population for this study was all women who underwent CS at the public hospitals in Harar city. Those women for whom CS procedure was conducted in the specified period are considered as our study population.

## Inclusion criteria and exclusion criteria

We reviewed all the medical records of women who underwent CS in the specified period. Medical records with incomplete values for the outcome and important predictor variables were excluded.

## Sample size

We conducted a census of all cesarean sections conducted in the two hospitals during the period of October 11, 2018 to December 31, 2019. Over this period, we found 1069 complete medical records documented with CS birth (806 records from HFSUH and 263 from Jugal General Hospital). We identified records of eligible women by reviewing hospital registry books in the operation room and at labor and obstetric wards.

## Data collection

Data collection checklists are developed after reviewing variables in patient's medical records. Five diploma midwives working at the study hospitals reviewed patient medical records and collected data using the data collection checklists. Two senior midwives, one at each hospital, supervised data collection process, and the principal investigator closely oversaw the overall data collection activity. The data collected included variables on characteristics related to demography, obstetrics, operation, comorbidity, and a post-CS SSI status. Age and residence are the only demographic characteristics for which values are collected. Data on obstetrics characteristics included parity (which refers to the number of pregnancies carried to fetal viability), gestational age (measured in weeks from last menstrual period to the date of CS), presence of labor before operation, duration of labor, membrane status before operation, duration of membrane rapture before operation, and chorioamnionitis (an inflammation of the fetal membranes, amnion and chorion, mostly due to bacterial infection). Regarding comorbidity data on cardiac disease, diabetes mellitus, hypertension, Human Immunodeficiency Virus/ Acquired Immunodeficiency Syndrome (HIV/AIDS), and anemia are collected. Variables collected to characterize the CS operation were the professional level of a physician conducting

the operation, operation type (elective versus emergency: a CS is emergency operation when it was done due to unexpected or acute obstetric emergencies, when the mother's or fetus's life or well-being is in direct jeopardy or risk [16]), duration of the operation, type of anesthesia (regional versus general), prophylactic antibiotic, duration of the operation, pre-operation hematocrit count, blood transfusion, type of abdominal incision (lower transverse versus vertical), and post-operation hospital stay in days), and a post-CS SSI status which was defined according to the Center for Disease Control and Prevention's (CDC) standard [5]. Post-CS SSI was the outcome variable and measured using five items with a yes/no response. The items used include: fever≥38°C, purulent discharge from surgical site, at least one sign of inflammation (pain/tenderness, localized swelling, redness or heat), abscess, and diagnosed wound infection by a physician/surgeon. A post-CS SSI was defined if there was at least one 'yes' response to any of the five items within 30 days after operation [5].

## Statistical analysis

The data were entered into Epi-data version 3.1 and exported to Stata version 16 software for analysis. Descriptive data analysis was made using frequency and percentages. Difference of proportion between two samples was conducted to see if there is a significant difference in the proportion of post-CS SSI between population groups. Bivariable and multivariable binary logistic regression analyses were conducted to identify factors associated with post-CS SSI. Variables that showed statistically significant association ($p < 0.25$) in the bivariable binary logistic regression are entered in the multivariable model. Model fitness of the final model was checked using the Hosmer-Lemeshow test in Stata (post-estimation command: estat gof) and it demonstrated a good fit with Pearson chi2 (133) = 149.7 and P-value = 0.153. Adjusted odds ratio (aOR) along with 95% CI was estimated to identify factors associated with post-CS SSI. Multicollinearity was checked using variance inflation factor (VIF) (higher VIF suggests possible existence of collinearity), and we removed a variable, 'presence of anemia', which showed collinearity with another variable, 'co-morbid conditions'. Statistical estimates were considered as significant at P-value < 0.05.

## Ethical consideration

This study obtained ethical approval from the Haramaya University, College of Health and Medical Sciences, Institutional Health Research Ethics Review Committee (IHRERC) with a reference number of IHRERC/016/2020. Formal letters of permission were written from the College to administrators of both Jugol General Hospital and HFSUH. Data collection was started after obtaining informed, written, voluntary and signed consent from the medical directors of the two hospitals involved. The ethics review committee waived obtaining informed consent from patients to access data. We collected the data anonymously without patient identifying information to maintain confidentiality of patient information.

## Results

### Demographic and obstetric characteristics

We reviewed complete medical record charts of 1069 women who underwent CS. The mean age (SD, standard deviation) was 27.4 (±5.3) years with 65.5% under the age of 30 years and 63.0% were rural residents. Regarding parity, 20.1%, 46.3%, and 21.3% were primi-para, multipara and grand-multipara, respectively. A significant number of women, 41.2%, presented to health facilities at gestational age < 37weeks, 21.7% had a rupture of membrane (ROM) before

operation, and 23.2% had Chorioamnionitis. From the total charts reviewed, 10.5% women had one co-morbid condition and 2.7% had at least two co-morbid conditions (Table 1).

## Operational and comorbidity characteristics

Emergency-CS was conducted for 75.9% of women and, a transverse abdominal incision was performed for 79.6% of women. Regarding comorbidity, 13.2% of women had at least one comorbid condition including cardiac illness, diabetes mellitus, hypertension, HIV/AIDS, and history of anemia. Only 24.0% of the operations were performed by a gynecologist and obstetrician with the remaining being conducted by residents from R1 to R4 (general practitioners specializing in gynecology and obstetrics, from year 1 to 4). The mean duration of the operations was 64 minutes (±16.1). Almost all, 95.2%, of the women received a prophylactic antibiotics before operation (Table 2).

**Table 1. The demographic and obstetric characteristics of women for whom cesarean section was conducted in two public hospitals in Harar city, East Ethiopia.**

| Variable | | Frequency | Percent |
|---|---|---|---|
| Demographic variables | | | |
| Age | <30 years old | 701 | 65.6 |
| | ≥30 years old | 368 | 34.4 |
| Residence | Urban | 395 | 37.0 |
| | Rural | 674 | 63.0 |
| Obstetric characteristics | | | |
| Parity | Nullipara | 131 | 12.3 |
| | Primipara | 215 | 20.1 |
| | Multipara | 495 | 46.3 |
| | Grand multipara | 228 | 21.3 |
| Gestational age | <37 weeks | 440 | 41.2 |
| | ≥ 37weeks | 629 | 58.8 |
| Labor before operation | Yes | 426 | 39.9 |
| | No | 643 | 60.1 |
| Status of membrane | Ruptured | 232 | 21.7 |
| | Intact | 837 | 78.3 |
| Chorioamnionitis | Yes | 248 | 23.2 |
| | No | 821 | 76.8 |
| Co-morbid conditions | | | |
| Cardiac disease | Yes | 12 | 1.1 |
| | No | 1057 | 98.9 |
| Diabetes | Yes | 24 | 2.3 |
| | No | 1045 | 97.7 |
| Hypertension | Yes | 28 | 2.6 |
| | No | 1041 | 97.4 |
| HIV | Yes | 18 | 1.7 |
| | No | 1051 | 98.3 |
| Anemia | Yes | 117 | 10.9 |
| | No | 952 | 89.1 |
| Overall comorbid conditions | No co-morbid condition | 928 | 86.8 |
| | one co-morbid condition | 112 | 10.5 |
| | two co-morbid conditions | 12 | 1.1 |
| | Three co-morbid conditions | 5 | 0.5 |
| | Four co-morbid conditions | 12 | 1.1 |

**Table 2. Operational characteristics women for whom cesarean section was conducted in two public hospitals in Harar city, East Ethiopia.**

| Variable | | Frequency | Percent |
|---|---|---|---|
| Type of operation | Elective | 258 | 24.1 |
| | Emergency | 811 | 75.9 |
| Type of Abdominal incision | Vertical | 218 | 20.4 |
| | Transverse | 851 | 79.6 |
| Post-operative hospital stay | <7days | 558 | 52.2 |
| | ≥7 day | 511 | 47.8 |
| Type of Anesthesia | General | 293 | 27.4 |
| | Regional | 776 | 72.6 |
| Who performed the operation? | Gynecologist | 256 | 24.0 |
| | Resident 4 | 306 | 28.6 |
| | ≤ Resident 3 | 507 | 47.4 |
| Blood transfusion | Yes | 134 | 12.5 |
| | No | 935 | 87.5 |
| Duration of operation | <60 minutes | 225 | 21.0 |
| | ≥60minutes | 844 | 79,0 |
| Anti-biotic prophylaxis | Yes | 1018 | 95.2 |
| | No | 51 | 4.8 |
| Pre-operative hematocrit count | <30% | 448 | 41.9 |
| | ≥ 30% | 621 | 58.1 |
| Anemia history | Yes | 117 | 10.9 |
| | No | 952 | 89.1 |

## Magnitude of post-CS SSI

The magnitude of post-CS SSI was 12.3% (95% CI: 10.4, 14.4) and tested to be significantly higher than a reference point estimate. A two tailed test for the difference between two proportions also indicated a positive and significantly a higher proportion of post-CS SSI among women who had raptured membrane, general anesthesia, more than 7 days hospital stay, and had chorioamnionitis. A higher proportion of post-CS SSI, 17.0% (95% CI: 10.8, 23.2), was observed among women with co-morbid conditions though the difference was not significant when compared with the proportion among women without co-morbid conditions. Furthermore, post-CS SSI was common among women who received a general anesthesia, 18.4% (95% CI: 14.0, 22.9) compared to women that received a regional anesthesia, 9.9% (95% CI: 7.8, 12.0) (Table 3).

Among the 13.2% of women with co-morbid conditions 10.9% had anemia, 2.6% were hypertensive, 2.3% were diabetics, 1.7% were HIV positive, and 1.1% had cardiac health problem. Stratifying the co-morbid conditions by women's hospital stay showed a significant association of women with no co-morbid conditions but stayed 7 days or more in hospital during postop and SSI (S1 Table).

## Factors associated with post-CS SSI

The following variables, age, parity, labor on presentation to hospital, membrane status, prophylactic antibiotics, chorioamnionitis, anemia, type of anesthesia, blood transfusion, post-operative hospital stay, comorbid conditions, and operation type are entered into the bi-variable binary logistic regression model. Variables with P-values <0.25 in the bi-variable model are considered in the multivariable binary logistic regression model. Four variables in the model including type of anesthesia, membrane status, post-operative hospital stay, and blood transfusion were significantly and positively associated with post-CS SSI. Accordingly, a two-

**Table 3. Post-CS SSI by the characteristics present before or during cesarean section procedures in two public hospitals in Harar city, East Ethiopia.**

| Variables | | Post-CS SSI, % (95% CI) | Standard error | P-value [‡] |
|---|---|---|---|---|
| Total sample (N = 1069) | | 12.3 (10.3, 14.4) | 0.010 | 0.005[¥] |
| Membrane status | Intact (n = 837) | 10.0 (8.0, 12.1) | .010 | <0.0001 |
| | Ruptured (n = 232) | 20.3 (15.1, 25.4) | .026 | |
| Operation type | Elective (n = 258) | 10.1 (6.4, 13.7) | 0.019 | 0.221 |
| | Emergency (n = 811) | 12.9 (10.6, 15.3) | 0.012 | |
| Co-morbid conditions | No (n = 928) | 11.5 (9.5, 13.6) | 0.010 | 0.064 |
| | Yes (n = 141) | 17.0 (10.8, 23.2) | 0.032 | |
| Anesthesia type | Regional (n = 776) | 9.9 (7.8, 12.0) | 0.011 | <0.0001 |
| | General (n = 293) | 18.4 (14.0, 22.9) | 0.023 | |
| Chorioamnionitis | No (n = 821) | 11.1 (8.9, 13.2) | 0.011 | 0.034 |
| | Yes (n = 248) | 16.1 (11.6, 20.7) | 0.023 | |
| Labor on presentation | No (n = 643) | 9.5 (7.2, 11.8) | 0.012 | 0.0007 |
| | Yes (n = 426) | 16.4 (12.9,20.0) | 0.018 | |
| Blood transfusion | No (n = 935) | 9.5 (7.6, 11.4) | 0.010 | <0.0001 |
| | Yes (n = 134) | 31.3 (23.5, 39.2) | 0.040 | |
| Hospital stay | <7 days (n = 558) | 7.3 (5.2, 9.5) | 0.011 | <0.0001 |
| | ≥ 7days (n = 511) | 17.6 (14.3, 20.9) | 0.017 | |

[¥] tested against 9.72% [17]

[‡] P-value (two sided) for independent sample test of proportions

fold increase in post-CS SSI was observed in women who received a general anesthesia (aOR = 2.02, 95% CI: 1.34, 3.02) and among those with ROM (aOR = 1.91, 95% CI: 1.18, 3.09). Similarly, women who stayed seven days or more in hospital after CS had over a two-fold increase in post-CS SSI compared to those who stayed less than seven days (aOR = 2.42, 95% CI: 1.61,3.64). More strong association with post-CS SSI was observed among women who had blood transfusion (aOR = 4.10, 95% CI: 2.61, 6.44) and had hospital stay over seven days (aOR = 2.42, 95% CI: 1.61, 3.64) (Table 4).

## Discussion

The magnitude of Post-CS SSI in this study was 12.3% (95%CI: 10.4, 14.4). General anesthesia, ROM before CS, more than 7 days post-operative hospital stay, and blood transfusion were found to independently and significantly increase post-CS SSI.

The post-CS SSI in this study, 12.3%, was higher compared to studies conducted in similar settings in Ethiopia and in other sub-Saharan countries that reported a magnitude of post-CS SSI as low as 4.1 to 8.4% [10, 18–21]. There are also studies that reported a magnitude that was consistent to the level reported in our study. For example, studies conducted in referral teaching hospitals in Ethiopia reported a magnitude of 11.4% in Jimma [22] and 11.0% in Hawassa [12]. However, a pooled analysis of post-CS SSI in sub-Saharan Africa reported a magnitude that was even higher amounting to 15.6% [23]. Variations in the magnitude of SSIs reported among studies may be due to a difference in the setting and the context in which the procedure was conducted, patient characteristics before and during operation, and surveillance of patients after surgery about wound healing or experience of complications. For example, in the current study, 22% of mothers had ROM before operation and 23% had chorioamnionitis. Moreover, 76% of the CS operations were on emergency basis. It was reported that these factors are shown to have positive association with risk of SSI development [24, 25].

**Table 4. Factors associated with post cesarean section surgical site infection in multivariable binary logistic regression model.**

| Variables | | cOR (95%CI) | aOR (95% CI)[†] |
|---|---|---|---|
| Labor present before operation | No | 1.00 | 1.00 |
| | Yes | 1.88 (1.30,2.71)** | 1.42(0.92, 2.19) |
| Chorioamnionitis | No | 1.00 | 1.00 |
| | Yes | 1.54 (1.03,2.31)* | 1.02(0.64,1.61) |
| Co-morbid conditions | No | 1.00 | 1.00 |
| | Yes | 1.57(0.97,2.55) | 1.56(0.93,2.61) |
| Type of Anesthesia | Regional | 1.00 | 1.00 |
| | General | 2.05(1.41,2.99)*** | 2.02 (1.34, 3.02)** |
| Status of Membrane | Intact | 1.00 | 1.00 |
| | Ruptured | 2.28(1.54,3.37)*** | 1.91(1.18,3.09)** |
| Post-operative hospital stay | <7 days | 1.00 | 1.00 |
| | ≥ 7days | 2.70(1.82,3.99)*** | 2.42 (1.61,3.64)*** |
| Blood transfusion | No | 1.00 | 1.00 |
| | Yes | 4.34(2.84,6.64)*** | 4.10(2.61,6.44)*** |
| Operation type | Elective | 1.00 | 1.00 |
| | Emergency | 1.33 (0.84, 2.09) | 1.29 (0.79, 2.10) |

* P-value < 0.05,

**P-value<0.01,

***P-value<0.0001, cOR = Crude odds ratio, and aOR = Adjusted odds ratio; [†]Hosmer-Lemeshow test, Pearson chi2 (133) = 149.7 and P-value = 0.153;

[†]mean variance inflation factor = 1.58.

Compared to the magnitude in more advanced settings where SSI rate was well below 5% [26–28], a higher magnitude of post-CS SSI is reported in settings where the magnitude of maternal mortality was the highest and the CS-rate was at a lower level [29, 30]. Research evidence indicated that SSIs are associated with a maternal mortality rate of up to 3% [31]. Coupled with an increasing trend in cesarean sections [30], particularly procedures conducted without an adequate indication, the problem of SSI needs well thought measures to prevent avoidable maternal deaths.

We found that women who received a general anesthesia for CS operation had twice the risk of SSI compared to those who received a regional anesthesia. This finding is consistent to reports from a large population-based study among Taiwanese women that reported a four-fold increase in post-CS SSI among those who received a general anesthesia [32]. Another large-scale population-based study in Japan reported an increase of sepsis and severe maternal morbidity among women who received a general anesthesia [33]. Furthermore, a meta-analysis on the prevalence and factors associated with post-CS SSI in Ethiopia reported that women who received a general anesthesia had over a two-fold increase in SSI [9]. A Cochrane review by Afolabi et al reported that general anesthesia is associated with an increase in maternal blood loss and hence maternal anemia, which in turn complicates wound healing [34].

Consistent with reports from different settings ROM before CS is positively and significantly associated with SSI. A meta-analysis on the association between membrane rapture and SSI reported a six-fold increase in post-CS SSI [35]. A case control study in Sierra Leone also reported a 50% increase in SSI among women with premature ROM [36]. These could be associated to the fact that a break in the sterility of the uterine cavity may give opportunity to an ascending infection and hence SSI [21].

In this study, we also found that post-operative hospital stay of seven days or more increased post-CS SSI. This finding is supported by research evidences reported from different

settings [37, 38]. Patients admitted for a long duration after CS are at risk of a nosocomial infections and also these women may have some condition that led them to stay long in hospital which in turn may be associated with delayed wound healing. However, in this study, we did not find a significant difference in the prevalence of post-CS SSI between women diagnosed with co-morbid conditions and those without. Consequently, we had no sufficient evidence (may be due to only a few women with co-morbid conditions) to report that existing co-morbid conditions contributed to patient's length of hospital stay.

The finding that perioperative blood transfusion was associated with post-CS SSI is also supported by findings from previous studies in Ethiopia and in other similar settings in Africa [22, 39]. This may be as a result of uncorrected anemia after transfusion for heavy blood loss. However, currently there is no recommendation to withhold necessary blood products from surgical patients as a means of either incisional or organ/space SSI risk reduction [22, 40].

## Strengths and limitations of the study

The strength of this study is that the findings are based on a census of hospital records of women who gave birth by CS in two big public hospitals in the study setting increasing precision of study estimates. However, since the hospitals did not have a surveillance system to track patients for whom CS procedure was conducted, estimates of the reported post-CS SSI could be an underestimate of magnitude of the problem. Furthermore, because data were collected from record reviews, important confounding variables (vaginal cleansing before CS, body mass index, indications for CS, types of antibiotics used for prophylaxis, microbiology of SSI, mortality, and estimated blood loss) might have been missed, and hence the reported results should be interpreted with caution. Finally, the design related limitation is that we cannot establish a temporal sequence for a cause-effect relationship.

## Conclusion

The post-CS SSI was high in public hospitals in Harar city. Types of anesthesia used, status of membrane, longer post-operative hospital stay, and blood transfusion were significantly associated with post-CS SSI. Hence, selection of anesthesia types, avoiding unnecessary prolonged hospitalization, and careful assessment of membrane status should be considered to avoid preventable SSI and maternal morbidity.

## Supporting information

**S1 Table. Surgical site infection by hospital stay after cesarean section operation and presence of comorbid conditions.**
(DOCX)

**S1 File. Data.**
(DTA)

## Author Contributions

**Conceptualization:** Tsegaw Alemye, Lemessa Oljira, Melkamu Merid Mengesha.

**Data curation:** Tsegaw Alemye.

**Formal analysis:** Tsegaw Alemye, Melkamu Merid Mengesha.

**Funding acquisition:** Tsegaw Alemye.

**Investigation:** Tsegaw Alemye.

**Methodology:** Tsegaw Alemye, Lemessa Oljira, Gelana Fekadu, Melkamu Merid Mengesha.

**Project administration:** Tsegaw Alemye.

**Resources:** Tsegaw Alemye.

**Supervision:** Tsegaw Alemye, Lemessa Oljira, Gelana Fekadu, Melkamu Merid Mengesha.

**Writing – original draft:** Tsegaw Alemye, Melkamu Merid Mengesha.

**Writing – review & editing:** Tsegaw Alemye, Lemessa Oljira, Gelana Fekadu, Melkamu Merid Mengesha.

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
