## [Decision Letter · Decision Letter 0]

18 Mar 2021

PONE-D-21-02570

Post Cesarean Section Surgical Site Infection and Associated Factors: A Hospital-based Analytic Cross-sectional Study

PLOS ONE

Dear Dr. Mengesha,

Thank you for submitting your manuscript to PLOS ONE. After careful consideration, we feel that it has merit but does not fully meet PLOS ONE’s publication criteria as it currently stands. Therefore, we invite you to submit a revised version of the manuscript that addresses the points raised during the review process.

We look forward to receiving your revised manuscript.

Kind regards,

Tze Shien Lo, MD

Academic Editor

PLOS ONE

Journal Requirements:

2. In the ethics statement in the manuscript and in the online submission form, please provide additional information about the patient records used in your retrospective study, including:

a) whether all data were fully anonymized before you accessed them;

b) the date range (month and year) during which patients' medical records were accessed;

c) the date range (month and year) during which patients whose medical records were selected for this study sought treatment.

If the ethics committee waived the need for informed consent, or patients provided informed written consent to have data from their medical records used in research, please include this information.

Additional Editor Comments:

1.Some grammatical errors noted in the manuscript and some sentences were not written in an intelligible fashion. As pointed out by reviewer 2, "Please also check for grammatical errors and ensure optimal flow of discussion for better understanding of study".

2. Line 120: Can the authors mention whether the residents from R1-R4 were being supervised by senior staff obstetricians when they performed CS?

3. Lines 197-198: The authors wrote "In this study, however, we did not collect data on whether vaginal preparation with antiseptic solution was done before CS......." So the authors need to explain why they could conclude "Appropriate vaginal cleansing with anti-septic solution , ....................should be considered to avoid preventable SSI and maternal mortality" (lines 220-222). The conclusion looks disorganized, it should be rewritten.

4. Please read the comments written by Reviewer 1 and Reviewer 2 and respond accordingly.

Reviewers' comments:

Reviewer's Responses to Questions

**Comments to the Author**

1. Is the manuscript technically sound, and do the data support the conclusions?

Reviewer #1: Partly

Reviewer #2: Partly

2. Has the statistical analysis been performed appropriately and rigorously? 

Reviewer #1: Yes

Reviewer #2: Yes

3. Have the authors made all data underlying the findings in their manuscript fully available?

Reviewer #1: Yes

Reviewer #2: Yes

4. Is the manuscript presented in an intelligible fashion and written in standard English?

Reviewer #1: Yes

Reviewer #2: No

5. Review Comments to the Author

Reviewer #1: Manuscript Number: PONE-D-21-02570

Manuscript Title: Post Cesarean Section Surgical Site Infection and Associated Factors: A Hospital-based Analytic Cross-sectional Study

This is a retrospective chart-review descriptive study which investigates variables associated with caesarian section (CS) post-surgical site infections (SSI) in two urban/rural teaching hospitals in Eastern Ethiopia. The Authors collected data from over 1000 CS during a 2-year period. By five criteria to define SSI, 12.5% of mothers developed infection. The Authors developed a binary logistic regression model to quantify the odds for individual variables relative to an odds ratio of 1.0 in participants lacking that variable. They identified four statistically significant risk factors for SSI and discussed these associations in their population. The methods used, the analysis and its presentation, and the discussion are strengths of the manuscript. They recognize limitations of the study. The results may be helpful for planning prospective observational controlled studies in the setting of developing African countries, which in turn may improve management to reduce morbidity associated with CS.

The following comments, which may improve the quality of the manuscript, should be addressed:

1) Mortality is not mentioned. How many maternal and newborn fatalities? Was this analyzed as a variable in the model and if so, was SSI associated with mortality?

2) There is no mention of the microbiology of the SSIs. If no data is available this should be made clear.

3) Page 5, line 50: How many were excluded from the sample?

4) Page 14, paragraph 2: Length of stay could be a result of SSI, or prolonged stay could result from some other condition and then predispose to nosocomial SSI. Was this point examined in the study and if not, it should be acknowledged that either possibility could be the case?

5) Lines 205 – 209: It is not clear whether anemia leading to transfusion was due to pre-existing etiologies and/or peripartum hemorrhage. Can the Authors shed light on this issue in the Discussion section?

6) Although not a variable in the analysis, the practice of “vaginal cleansing” is mentioned and referenced as standard management for prevention of postpartum infections. It seems possible that quite the opposite may be true. That is eradication of normal vaginal flora by alcohol-iodine disinfectant could predispose to endometritis and SSI. Are the Authors on solid ground regarding their statements on this issue (lines 27 -28; line 220)? Is this a practice to prevent maternal to newborn transmission of HIV-1?

7) Line 52: Should not this be 24 months (instead of 12 months)?

8) The antibiotics used for surgical prophylaxis are not mentioned.

9) Lines 127 – 128: What is this reference point estimate and from where is it derived?

10) With the high prevalence of emergency CS (76%), it would be helpful for the reader to know whether this affected timely antibiotic prophylaxis and disinfection for operation. Admittedly this may be difficult to glean from the hospital records retrospectively.

Reviewer #2: A strength of the study is that it gives us an idea of the state of surgical site infection in this part of eastern Africa.

Please address several clarifications.

On the abstract, please revise the aim of the study to specify the population being studied. Also, please revise the abstract conclusion. A prevalence of "very high" is a very subjective and strong conclusion to make that is not supported by the results.

On introduction (1st paragraph), describe the population with post-CS SSI cumulative incidence of 11.8%. clearly, this is too high for developed countries. Please specify the population being described specifically on all data from literature review i.e. SSI carries 60% ICU stay, readmission and death.

4th paragraph, 2nd "sentence" is not a sentence.

5th paragraph in introduction, again, please note context of literature review that "despite infection prevention efforts, no significant decline is achieved".

Methods:

Study period is Jan 1, 2018 - Dec 2019. The March 2020 data collection timeframe is not the study period and misleading. Also it is very important to add more on the study setting. Please describe it more than the population region. We need to have a better understanding of healthcare system in the country/area so as to put the results into context.

Sample size: it is a census of 24 months and not just 12

How many total charts were reviewed and how many ended up on analysis. It would be best to put a diagram on how you ended up with 1069 and how many were excluded.

Results; Do you have any data on indication for CS? Are the hospitals performing more CS than expected thus resulting in more SSI? Are the hospitals referral centers seeing more complicated cases thus requiring more CS? I think this data plus indications would make the study stronger.

You also have data on comorbid conditions. These should be included in table 1 with each prevalence data separated out per comorbid condition.

What is the standard preop antibiotics used in the hospitals?

Conclusion: Please revise conclusion to align with results. No data on vaginal antiseptic but was included in conclusion. Also the antibiotics was not significant to be in there as well.

Please also check for grammatical errors and ensure optimal flow of discussion for better understanding of study.

6. PLOS authors have the option to publish the peer review history of their article (what does this mean?). If published, this will include your full peer review and any attached files.

Reviewer #1: No

Reviewer #2: No

---

## [Author Response · Author response to Decision Letter 0]

2 May 2021

RESPONSE TO REVIEWERS

Comment from the editor and authors’ response

and

o Thank you for providing these helpful links.

 2. In the ethics statement in the manuscript and in the online submission form, please provide additional information about the patient records used in your retrospective study, including:

a. whether all data were fully anonymized before you accessed them;

o We collected data from the operation log books and patient card. Patient self-identifying information are recorded in these documents. However, we collected the data anonymously using patient medical record number. 

b) The date range (month and year) during which patients' medical records were accessed;

o Data were accessed from March 1-15, 2020. This can be accessed in line 30 of the revised manuscript.

c) The date range (month and year) during which patients whose medical records were selected for this study sought treatment.

o October 11, 2018 to December 31st 2019. This can accessed in line 29 of the revised manuscript.

If the ethics committee waived the need for informed consent, or patients provided informed written consent to have data from their medical records used in research, please include this information.

o Thank you for the helpful suggestion. We have now included a statement describing this in the revised manuscript. This information can be obtained in lines 104-110.

Additional Editor Comments: 

1.Some grammatical errors noted in the manuscript and some sentences were not written in an intelligible fashion. As pointed out by reviewer 2, "Please also check for grammatical errors and ensure optimal flow of discussion for better understanding of study".

o Thank you for the comments and suggestions. In the revised manuscript, we thoroughly checked grammars and the coherence of ideas throughout the document. 

2. Line 120: Can the authors mention whether the residents from R1-R4 were being supervised by senior staff obstetricians when they performed CS?

o We thank the editor for raising this concern. Though the authors couldn’t get these data from patient records, it can be stated that senior staff supervise residents as part of performance evaluation.

3. Lines 197-198: The authors wrote "In this study, however, we did not collect data on whether vaginal preparation with antiseptic solution was done before CS......." So the authors need to explain why they could conclude "Appropriate vaginal cleansing with anti-septic solution, ....................should be considered to avoid preventable SSI and maternal mortality" (lines 220-222). The conclusion looks disorganized, it should be rewritten.

o We thank the editor for the comment and suggestion to avoid statements in the conclusion that are not supported by study findings. The authors appreciate this concern and corrected the statement in the revised manuscript. 

4. Please read the comments written by Reviewer 1 and Reviewer 2 and respond accordingly.

Review Comments to the Author

Reviewer #1: Manuscript Number: PONE-D-21-02570

Manuscript Title: Post Cesarean Section Surgical Site Infection and Associated Factors: A Hospital-based Analytic Cross-sectional Study

This is a retrospective chart-review descriptive study which investigates variables associated with caesarian section (CS) post-surgical site infections (SSI) in two urban/rural teaching hospitals in Eastern Ethiopia. The Authors collected data from over 1000 CS during a 2-year period. By five criteria to define SSI, 12.5% of mothers developed infection. The Authors developed a binary logistic regression model to quantify the odds for individual variables relative to an odds ratio of 1.0 in participants lacking that variable. They identified four statistically significant risk factors for SSI and discussed these associations in their population. The methods used, the analysis and its presentation, and the discussion are strengths of the manuscript. They recognize limitations of the study. The results may be helpful for planning prospective observational controlled studies in the setting of developing African countries, which in turn may improve management to reduce morbidity associated with CS.

The following comments, which may improve the quality of the manuscript, should be addressed: 

1) Mortality is not mentioned. How many maternal and newborn fatalities? Was this analyzed as a variable in the model and if so, was SSI associated with mortality?

o Thank you for the concerns. As the variables, maternal and newborn mortality, were not part of the study objectives, we did not collect data on these variables. The suggestion on whether association existed between SSI and mortality was very interesting. However, we did not collect data on this and reported as limitation of our study in the revised manuscript in lines 231-33.

2) There is no mention of the microbiology of the SSIs. If no data is available this should be made clear.

o Thank for raising this concern. As we reviewed medical record chart, we could not found data concerning the microbiology of SSI. Hence, we reported this as a limitation in the revised manuscript in lines 231-33.

3) Page 5, line 50: How many were excluded from the sample?

o Sorry for the editorial error in the previous submission in line 106 (result: sociodemographic section). Because we did a census, we included 1069 complete medical record reviews and there was no excluded record from analysis. 

4) Page 14, paragraph 2: Length of stay could be a result of SSI, or prolonged stay could result from some other condition and then predispose to nosocomial SSI. Was this point examined in the study and if not, it should be acknowledged that either possibility could be the case?

o Thank you for raising this important concern. We did not see the data in this way before the reviewer’s suggestion. We presented how hospital stay and comorbid conditions are related with surgical site infection in Table 3 of the original submission. To address the reviewer’s concern, we stratified occurrence of surgical site infection by the length of hospital stay and presence of comorbid conditions. The results indicated that women with no comorbid conditions but with hospital stay of more than seven days had a higher magnitude of surgical site infection. These findings are presented in a supplementary table S1 in the revised manuscript in lines 152-154.

o 

5) Lines 205 – 209: It is not clear whether anemia leading to transfusion was due to pre-existing etiologies and/or peripartum hemorrhage. Can the Authors shed light on this issue in the Discussion section?

o This is very interesting point. However, we have no data on the cause of anemia. In the revised manuscript, we added details in the discussion section with regard to how anemia contributes post-surgical site infection. Furthermore, in the limitation section, we mentioned that we did not have the data on what caused the anemia. This can be found in the revised manuscript in lines 231-33.

6) Although not a variable in the analysis, the practice of “vaginal cleansing” is mentioned and referenced as standard management for prevention of postpartum infections. It seems possible that quite the opposite may be true. That is eradication of normal vaginal flora by alcohol-iodine disinfectant could predispose to endometritis and SSI. Are the Authors on solid ground regarding their statements on this issue (lines 27 -28; line 220)? Is this a practice to prevent maternal to newborn transmission of HIV-1?

o Of course, evidence may be inconclusive in this regard, and we share the reviewer’s concern. A Cochrane Review of 21 randomized controlled trials which participated a total of 7038 women undergoing cesarean section compared the infection rate when vaginal cleansing (povidone-iodine or chlorhexidine) is used and not used. The findings indicated that reduced infection was observed among the intervention groups for whom either povidone-iodine or chlorhexidine is used (reference: Haas DM, Morgan S, Contreras K, Kimball S. Vaginal preparation with antiseptic solution before cesarean section for preventing postoperative infections. Cochrane Database of Systematic Reviews 2020, Issue 4. Art. No.: CD007892. DOI: 10.1002/14651858.CD007892.pub7). 

o Because we did not collect data on the use of vaginal cleansing in the study setting, in the revised manuscript we removed the statement presented in the conclusion regarding vaginal cleansing. 

7) Line 52: Should not this be 24 months (instead of 12 months)?

o Thank you for the suggestion. We corrected this in the revised manuscript in lines 29-30.

8) The antibiotics used for surgical prophylaxis are not mentioned.

o Thank you for the concern. Of course, we reported whether or not antibiotics was given preoperative. Consequently, we have no data to report with regard to specific antibiotics used. Indicating the type antibiotics used might have shaded light how effectively patients are protected from SSI. 

o As general information, however, Ceftriaxone 1gm IV stat and Ampicillin 1gm IV TID for one day are commonly used in the hospitals studied. 

9) Lines 127 – 128: What is this reference point estimate and from where is it derived?

o Thank you for the question asked for clarity. We conducted a one-sample test of proportion (the general syntax in Stata for one sample test of proportions is given by prtest varname == #p, [options]; for a two sample test, prtest varname [if] [in], by (groupvar)) for surgical site infection in our study sample by comparing it against a hypothesized population value. The value for the population parameter, for the particular question asked, is obtained from a meta-analysis which was conducted in Ethiopia and published in 2020 (Getaneh, T., Negesse, A. & Dessie, G. Prevalence of surgical site infection and its associated factors after cesarean section in Ethiopia: systematic review and meta-analysis. BMC Pregnancy Childbirth 20, 311 (2020). https://doi.org/10.1186/s12884-020-03005-8).

10) With the high prevalence of emergency CS (76%), it would be helpful for the reader to know whether this affected timely antibiotic prophylaxis and disinfection for operation. Admittedly this may be difficult to glean from the hospital records retrospectively.

o As the reviewer mentioned, we did not collect data on whether the high burden of emergency CS affected antibiotic administration and also disinfection for operation.

Reviewer 2:

Reviewer #2: A strength of the study is that it gives us an idea of the state of surgical site infection in this part of eastern Africa.

Please address several clarifications.

On the abstract, please revise the aim of the study to specify the population being studied. Also, please revise the abstract conclusion. A prevalence of "very high" is a very subjective and strong conclusion to make that is not supported by the results.

o Thank you for the comment and we corrected these in the revised manuscript. 

On introduction (1st paragraph), describe the population with post-CS SSI cumulative incidence of 11.8%. Clearly, this is too high for developed countries. Please specify the population being described specifically on all data from literature review i.e. SSI carries 60% ICU stay, readmission and death.

Response: 

o Thank you for the comments. 

o The 11.8% magnitude of Post-CS SSI is the figure in low and middle income countries. We have corrected and specified the population as commented in the revised manuscript under the introduction section, Paragraph II, line 10-12. 

o With regard to the statement “SSI carries 60% ICU stay, readmission and death,” our intention is to show the time a women with SSI spend in ICU as compared to those without SSI. Otherwise, it is not about the magnitude of SSI in ICU. We removed this statement in the revised manuscript.

4th paragraph, 2nd "sentence" is not a sentence.

o Thank you for catching this. We rewrote it in the revised manuscript as “Research evidences reported from different settings indicated that several factors contribute to SSIs including a preexisting comorbid conditions, age, duration of operation, anemia, frequency of manual vaginal examination, and inappropriate antibiotic prophylaxis (13-15). Hence, identification of such risk factors that increase SSIs may help to design interventions that consider the context where such operations are performed. These changes can be found in lines 16-20 in the revised manuscript.

5th paragraph in introduction, again, please note context of literature review that "despite infection prevention efforts, no significant decline is achieved".

o Thank you for the suggestion. We acknowledged it and corrected in the revised manuscript in lines 23-24.

Methods: 

Study period is Jan 1, 2018 - Dec 2019. The March 2020 data collection timeframe is not the study period and misleading. Also it is very important to add more on the study setting. Please describe it more than the population region. We need to have a better understanding of healthcare system in the country/area so as to put the results into context.

o Thank for the comments and suggestions. We acknowledged both the comments and suggestions and corrected them in the revised manuscript. We have also added details about the Ethiopia’s healthcare structure and distribution of facilities together with the populations served in this section. The changes made can be found in lines 34-43 in the revised manuscript.

Sample size: it is a census of 24 months and not just 12

o Thank you for the comment. We acknowledged the comment and corrected it in the revised manuscript. To collect a complete year data both in the Ethiopian and the Gregorian calendar, we collected 15 months of data, beginning from October 2018 to December 2019. The new year in the Ethiopian calendar begins in September; we collected data from the new year beginning (2011 E.C) in the Ethiopian calendar, which corresponds to September 2018, to the last month of the next year in the Gregorian calendar, December 2019. Accordingly, the total months our study period is 15 months. Sorry for the misleading texts provided in the previous manuscript.

How many total charts were reviewed and how many ended up on analysis. It would be best to put a diagram on how you ended up with 1069 and how many were excluded.

o In the 15 months considered, between October 2018 to December 2019, we identified a total of 1069 CS deliveries from the two hospitals (806 records from Hiwot Fana Specialized University Hospital and 263 from Jugal Hospital). Consequently, we selected all the identified records and included in the analysis. 

Results; Do you have any data on indication for CS? Are the hospitals performing more CS than expected thus resulting in more SSI? Are the hospitals referral centers seeing more complicated cases thus requiring more CS? I think this data plus indications would make the study stronger.

o Thank you for the very helpful concerns regarding the request for CS indications and also the trends of CS and the types of cases visiting the study settings. However, we did not collect data on indications of CS as we considered them to be not within the scope of the study objective. Of course, there might be increase in the number of CS procedures as the young and affluent families prefer not to deliver by labor. It can be expected that referral hospitals often receive complicated cases as the availability of human and material resources allow them to handle such cases. However, beyond making a guess, we did not identify referral cases and also the reasons for referral and the nature of complications of referral cases. 

You also have data on comorbid conditions. These should be included in table 1 with each prevalence data separated out per comorbid condition.

o Thank you for the comment. We included the specific comorbid conditions and the total comorbid conditions as women may have more than one conditions in Table 1 as suggested.

What is the standard preop antibiotics used in the hospitals?

o Thank you for raising this question. As previously described in our response to the first reviewer’s question, however, we didn’t collect specific data about the type of antibiotics administered. Rather, we collected data on whether antibiotics was administered or not. As a general information to respond to the reviewers’ questions, there is no specific standard preoperative antibiotics in the hospitals in the study setting. The choice of antibiotics, hence, depends on the choice and decision of the physician. As general information, Ceftriaxone 1gm IV stat and Ampicillin 1gm IV TID for one day are commonly used in the hospitals studied. 

Conclusion: Please revise conclusion to align with results. No data on vaginal antiseptic but was included in conclusion. Also the antibiotics was not significant to be in there as well.

Please also check for grammatical errors and ensure optimal flow of discussion for better understanding of study.

o Response: We have accepted the comments and corrected accordingly. The conclusion section has been modified as “The post-CS SSI was high in public hospitals of Harar town. Types of anesthesia used, status of membrane, longer post-operative Hospital stay, and blood transfusion were significantly associated with post cesarean section surgical site infection. Correspondingly, selection of anesthesia types, avoiding unnecessary hospital delay, careful assessment of membrane status, and blood transfusion should be considered to avoid preventable SSI and maternal morbidity.” The changes made can be found in lines 238-242.

---

## [Editor Report · Decision Letter 1]

19 May 2021

PONE-D-21-02570R1

Post Cesarean Section Surgical Site Infection and Associated Factors among Women Who Delivered in Public Hospitals in Harar City, Eastern Ethiopia: A Hospital-based Analytic Cross-sectional Study

PLOS ONE

Dear Dr. Melkamu Merid,

Thank you for submitting your manuscript to PLOS ONE. After careful consideration, we feel that it has merit but does not fully meet PLOS ONE’s publication criteria as it currently stands. Therefore, we invite you to submit a revised version of the manuscript that addresses the points raised during the review process.

We look forward to receiving your revised manuscript.

Kind regards,

Tze Shien Lo, MD

Academic Editor

PLOS ONE

Journal Requirements:

Additional Editor Comments (if provided):

I would like to thank the authors for their great effort in significantly improved the quality of their manuscript by answering the comments made by the editor and reviewers. However, the authors need to make the following minor revisions before our acceptance of the manuscript.

1. In Abstract under Results line 4-5 '.......had at least one co-morbid conditions.' should be changed to '.......had at least one co-morbid condition.'

2. In abstract under Conclusion line 3 '......to avoid unnecessary delays in hospital......' does not make much sense. It should be rewritten as '.......to avoid unnecessary prolonged hospitalization.....'

3.Under Study area and Period line 29 'from October 11, 2018 and December 31st, 2019' should be changed to 'from October 11, 2018 to December 31, 2019'.

4. Under Study area and Period line 30 '....during March 1-15/2020 from two public.....'should be changed to '......during March 1 to March 15, 2020 at two public.....'.

5. Under Data collection line 87 '.......if there is at least one 'yes' response....' should be changed to '.......if there was at least one 'yes' response....'

6. Under Demographic and Obstetric Characteristics line 114 '......with most, 78.4%, being under the age of .....' should be rewritten to '......with 65.6% under the age of .....'. In table 1, the number is 65,6%, am i missing something? Please explain.

7. Under Strengths and limitations of the study line232 does BMI mean body mass index? if yes, they should be spelt out.

---

## [Author Response · Author response to Decision Letter 1]

24 May 2021

Additional Editor Comments (if provided):

I would like to thank the authors for their great effort in significantly improved the quality of their manuscript by answering the comments made by the editor and reviewers. However, the authors need to make the following minor revisions before our acceptance of the manuscript.

o We highly appreciate that the editor valued the changes we have made in the revised manuscript (PONE-D-21-02570R1) following the comments provided in the first revision. We thank the editor and reviewers for their time and valuable comments and suggestions that improved our manuscript. 

1. In Abstract under Results line 4-5 '.......had at least one co-morbid conditions.' should be changed to '.......had at least one co-morbid condition.'

o Thank you for the edits and the suggestion. We have now corrected it in the revised version as: “Emergency CS was conducted for 75.9% (95% CI: 73.2, 78.3) of the women and 13.2% (95% CI: 11.3, 15.4) had at least one co-morbid condition.” The changes can be found in lines 32-33 of the revised manuscript.

2. In abstract under Conclusion line 3 '......to avoid unnecessary delays in hospital......' does not make much sense. It should be rewritten as '.......to avoid unnecessary prolonged hospitalization.....'

o Thank you for the edits and the suggestion. We have now corrected it in the revised version as: “Screening for preoperative anemia and appropriate correction before surgery, selection of the type of anesthesia, close follow-up to avoid unnecessary prolonged hospitalization, and careful assessment of membrane status should be considered to avoid preventable SSI and maternal morbidity.” The changes can be found in lines 39-42 of the revised manuscript.

3.Under Study area and Period line 29 'from October 11, 2018 and December 31st, 2019' should be changed to 'from October 11, 2018 to December 31, 2019'.

o Thank you for the edits and the suggestion. We have now corrected it in the revised version as: “A fifteen months data, from October 11, 2018 to December 31, 2019, …” The changes can be found in line 72 of the revised manuscript.

4. Under Study area and Period line 30 '....during March 1-15/2020 from two public.....'should be changed to '......during March 1 to March 15, 2020 at two public.....'.

o Thank you for the edits and the suggestion. We have now corrected it in the revised version as: “….was extracted from patient records during March 1 to March 15, 2020 from two public hospitals in Harar city.” The changes can be found in line 73 of the revised manuscript.

5. Under Data collection line 87 '.......if there is at least one 'yes' response....' should be changed to '.......if there was at least one 'yes' response....'

o Thank you for the edits and the suggestion. We have now corrected it in the revised version as: “A post-CS SSI was defined if there was at least one ‘yes’ response to any of the five items within 30 days after operation.” The changes can be found in lines 129-130 of the revised manuscript.

6. Under Demographic and Obstetric Characteristics line 114 '......with most, 78.4%, being under the age of .....' should be rewritten to '......with 65.6% under the age of .....'. In table 1, the number is 65,6%, am i missing something? Please explain.

o Thank you for the edits and the suggestion. We have now corrected it in the revised version as: “The mean age (SD, standard deviation) was 27.4 (±5.3) years with 65.5% under the age of 30 years and 63.0% were rural residents.” The changes can be found in lines 156-158 of the revised manuscript.

7. Under Strengths and limitations of the study line232 does BMI mean body mass index? if yes, they should be spelt out.

o Thank you for the edits and the suggestion. We have now corrected it in the revised version as: “important confounding variables (vaginal cleansing before CS, body mass index, indications for CS…) might have been missed.” The changes can be found in lines 275 of the revised manuscript.

---

## [Editor Report · Decision Letter 2]

31 May 2021

Post Cesarean Section Surgical Site Infection and Associated Factors among Women Who Delivered in Public Hospitals in Harar City, Eastern Ethiopia: A Hospital-based Analytic Cross-sectional Study

PONE-D-21-02570R2

Dear Dr. Mengesha

We’re pleased to inform you that your manuscript has been judged scientifically suitable for publication and will be formally accepted for publication once it meets all outstanding technical requirements.

Kind regards,

Tze Shien Lo, MD

Academic Editor

PLOS ONE
---

## [Editor Report · Acceptance letter]

14 Jun 2021

PONE-D-21-02570R2 

Post Cesarean Section Surgical Site Infection and Associated Factors among Women Who Delivered in Public Hospitals in Harar City, Eastern Ethiopia: A Hospital-based Analytic Cross-sectional Study 

Dear Dr. Mengesha:

I'm pleased to inform you that your manuscript has been deemed suitable for publication in PLOS ONE. Congratulations! Your manuscript is now with our production department. 

Kind regards, 

on behalf of

Dr. Tze Shien Lo 

Academic Editor

PLOS ONE